# Towards a Unified View of Parameter-Efficient Transfer Learning

**Junxian He**[*]
Carnegie Mellon University
junxianh@cs.cmu.edu

**Chunting Zhou**[*]
Carnegie Mellon University
chuntinz@cs.cmu.edu

**Xuezhe Ma**
University of Southern California
xuezhema@isi.edu

**Taylor Berg-Kirkpatrick**
UC San Diego
tberg@eng.ucsd.edu

**Graham Neubig**
Carnegie Mellon University
gneubig@cs.cmu.edu

## Abstract

Fine-tuning large pretrained language models on downstream tasks has become the de-facto learning paradigm in NLP. However, conventional approaches fine-tune all the parameters of the pretrained model, which becomes prohibitive as the model size and the number of tasks grow. Recent work has proposed a variety of parameter-efficient transfer learning methods that only fine-tune a small number of (extra) parameters to attain strong performance. While effective, the critical ingredients for success and the connections among the various methods are poorly understood. In this paper, we break down the design of state-of-the-art parameter-efficient transfer learning methods and present a unified framework that establishes connections between them. Specifically, we re-frame them as modifications to specific hidden states in pretrained models, and define a set of design dimensions along which different methods vary, such as the function to compute the modification and the position to apply the modification. Through comprehensive empirical studies across machine translation, text summarization, language understanding, and text classification benchmarks, we utilize the unified view to identify important design choices in previous methods. Furthermore, our unified framework enables the transfer of design elements across different approaches, and as a result we are able to instantiate new parameter-efficient fine-tuning methods that tune less parameters than previous methods while being more effective, achieving comparable results to fine-tuning all parameters on all four tasks.[1]

## 1 Introduction

Transfer learning from pre-trained language models (PLMs) is now the prevalent paradigm in natural language processing, yielding strong performance on many tasks (Peters et al., 2018; Devlin et al., 2019; Qiu et al., 2020). The most common way to adapt general-purpose PLMs to downstream tasks is to fine-tune all the model parameters (*full fine-tuning*). However, this results in a separate copy of fine-tuned model parameters for each task, which is prohibitively expensive when serving models that perform a large number of tasks. This issue is particularly salient with the ever-increasing size of PLMs, which now range from hundreds of millions (Radford et al., 2019; Lewis et al., 2020) to hundreds of billions (Brown et al., 2020) or even trillions of parameters (Fedus et al., 2021).

To mitigate this issue, a few lightweight alternatives have been proposed to update only a small number of extra parameters while keeping most pretrained parameters frozen. For example, *adapter tuning* (Houlsby et al., 2019) inserts small neural modules called adapters to each layer of the pretrained network and only the adapters are trained at fine-tuning time. Inspired by the success of prompting methods that control PLMs through textual prompts (Brown et al., 2020; Liu et al., 2021a), *prefix tuning* (Li & Liang, 2021) and *prompt tuning* (Lester et al., 2021) prepend an additional $l$ tunable

---

[*]Equal Contribution. Order determined by random dice rolling.
[1]Code is available at https://github.com/jxhe/unify-parameter-efficient-tuning.

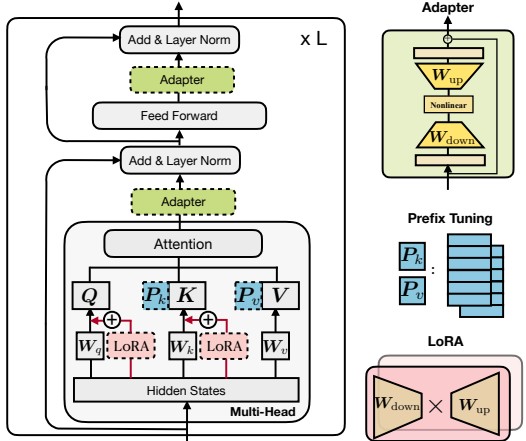

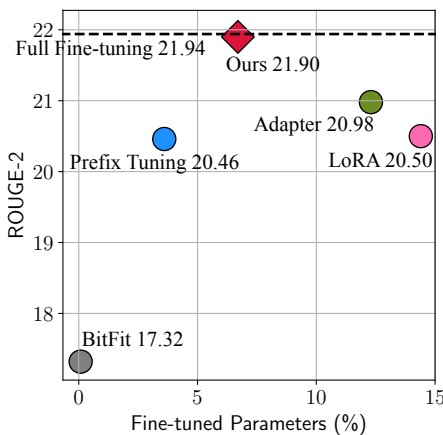

Figure 1: Illustration of the transformer architecture and several state-of-the-art parameter-efficient tuning methods. We use blocks with dashed borderlines to represent the added modules by those methods.

Figure 2: Performance of different methods on the XSum (Narayan et al., 2018) summarization task. The number of fine-tuned parameters is relative to the tuned parameters in full fine-tuning.

prefix tokens to the input or hidden layers and only train these soft prompts when fine-tuning on downstream tasks. More recently, Hu et al. (2021) learn low-rank matrices to approximate parameter updates. We illustrate these methods in Figure 1. These approaches have all been reported to demonstrate comparable performance to full fine-tuning on different sets of tasks, often through updating less than 1% of the original model parameters. Besides parameter savings, parameter-efficient tuning makes it possible to quickly adapt to new tasks without catastrophic forgetting (Pfeiffer et al., 2021) and often exhibits superior robustness in out-of-distribution evaluation (Li & Liang, 2021).

However, we contend that the important ingredients that contribute to the success of these parameter-efficient tuning methods are poorly understood, and the connections between them are still unclear. In this paper, we aim to answer three questions: (1) How are these methods connected? (2) Do these methods share design elements that are essential for their effectiveness, and what are they? (3) Can the effective ingredients of each method be transferred to others to yield more effective variants?

In order to answer these questions, we first derive an alternative form of prefix tuning that reveals prefix tuning's close connections with adapters (§3.1). Based on this we then devise a unified framework that frames the aforementioned methods as different ways to modify the hidden representations of frozen PLMs (§3.2). Our unified framework decomposes previous methods along a *shared* set of design dimensions, such as the function used to perform the modification, the position in which to impose this modification, and how to integrate the modification. This framework allows us to transfer design choices across approaches to propose new variants such as adapters with multiple heads (§3.3). In experiments, we first show that existing parameter-efficient tuning methods still lag behind full fine-tuning on higher-resource and challenging tasks (§4.2), as exemplified in Figure 2. Then we utilize the unified framework to identify critical design choices and validate the proposed variants empirically (§4.3-4.6). Our experiments on four NLP benchmarks covering text summarization, machine translation (MT), text classification, and general language understanding, demonstrate that the proposed variant uses less parameters than existing methods while being more effective, matching full fine-tuning results on all four tasks.

## 2 PRELIMINARIES

### 2.1 RECAP OF THE TRANSFORMER ARCHITECTURE

The transformer model (Vaswani et al., 2017) is now the workhorse architecture behind most state-of-the-art PLMs. In this section we recap the equations of this model for completeness. Transformer models are composed of $L$ stacked blocks, where each block (Figure 1) contains two types of sub-

layers: multi-head self-attention and a fully connected feed-forward network (FFN).[2] The conventional attention function maps queries $\boldsymbol{Q} \in \mathbb{R}^{n \times d_k}$ and key-value pairs $\boldsymbol{K} \in \mathbb{R}^{m \times d_k}, \boldsymbol{V} \in \mathbb{R}^{m \times d_v}$:

$$\text{Attn}(\boldsymbol{Q}, \boldsymbol{K}, \boldsymbol{V}) = \text{softmax}(\frac{\boldsymbol{Q}\boldsymbol{K}^T}{\sqrt{d_k}})\boldsymbol{V}, \tag{1}$$

where $n$ and $m$ are the number of queries and key-value pairs respectively. Multi-head attention performs the attention function in parallel over $N_h$ heads, where each head is separately parameterized by $\boldsymbol{W}_q^{(i)}, \boldsymbol{W}_k^{(i)}, \boldsymbol{W}_v^{(i)} \in \mathbb{R}^{d \times d_h}$ to project inputs to queries, keys, and values. Given a sequence of $m$ vectors $\boldsymbol{C} \in \mathbb{R}^{m \times d}$ over which we would like to perform attention and a query vector $\boldsymbol{x} \in \mathbb{R}^d$, multi-head attention (MHA) computes the output on each head and concatenates them:[3]

$$\text{MHA}(\boldsymbol{C}, \boldsymbol{x}) = \text{Concat}(\text{head}_1, \cdots, \text{head}_\text{h})\boldsymbol{W}_o, \ \text{head}_\text{i} = \text{Attn}(\boldsymbol{x}\boldsymbol{W}_q^{(i)}, \boldsymbol{C}\boldsymbol{W}_k^{(i)}, \boldsymbol{C}\boldsymbol{W}_v^{(i)}), \tag{2}$$

where $\boldsymbol{W}_o \in \mathbb{R}^{d \times d}$. $d$ is the model dimension, and in MHA $d_h$ is typically set to $d/N_h$ to save parameters, which indicates that each attention head is operating on a lower-dimensional space. The other important sublayer is the fully connected feed-forward network (FFN) which consists of two linear transformations with a ReLU activation function in between:

$$\text{FFN}(\boldsymbol{x}) = \text{ReLU}(\boldsymbol{x}\boldsymbol{W}_1 + \boldsymbol{b}_1)\boldsymbol{W}_2 + \boldsymbol{b}_2, \tag{3}$$

where $\boldsymbol{W}_1 \in \mathbb{R}^{d \times d_m}, \boldsymbol{W}_2 \in \mathbb{R}^{d_m \times d}$. Transformers typically use a large $d_m$, e.g. $d_m = 4d$. Finally, a residual connection is used followed by layer normalization (Ba et al., 2016).

## 2.2 Overview of Previous Parameter-efficient Tuning Methods

Below and in Figure 1, we introduce several state-of-the-art parameter-efficient tuning methods. Unless otherwise specified, they only tune the added parameters while the PLM's are frozen.

**Adapters (Houlsby et al., 2019):** The adapter approach inserts small modules (adapters) between transformer layers. The adapter layer generally uses a down-projection with $\boldsymbol{W}_\text{down} \in \mathbb{R}^{d \times r}$ to project the input $\boldsymbol{h}$ to a lower-dimensional space specified by bottleneck dimension $r$, followed by a nonlinear activation function $f(\cdot)$, and a up-projection with $\boldsymbol{W}_\text{up} \in \mathbb{R}^{r \times d}$. These adapters are surrounded by a residual connection, leading to a final form:

$$\boldsymbol{h} \leftarrow \boldsymbol{h} + f(\boldsymbol{h}\boldsymbol{W}_\text{down})\boldsymbol{W}_\text{up}. \tag{4}$$

Houlsby et al. (2019) places two adapters sequentially within one layer of the transformer, one after the multi-head attention and one after the FFN sub-layer. Pfeiffer et al. (2021) have proposed a more efficient adapter variant that is inserted only after the FFN "add & layer norm" sub-layer.

**Prefix Tuning (Li & Liang, 2021):** Inspired by the success of textual prompting methods (Liu et al., 2021a), prefix tuning prepends $l$ tunable prefix vectors to the keys and values of the multi-head attention at every layer. Specifically, two sets of prefix vectors $\boldsymbol{P}_k, \boldsymbol{P}_v \in \mathbb{R}^{l \times d}$ are concatenated with the original key $\boldsymbol{K}$ and value $\boldsymbol{V}$. Then multi-head attention is performed on the new prefixed keys and values. The computation of $\text{head}_i$ in Eq. 2 becomes:

$$\text{head}_i = \text{Attn}(\boldsymbol{x}\boldsymbol{W}_q^{(i)}, \text{concat}(\boldsymbol{P}_k^{(i)}, \boldsymbol{C}\boldsymbol{W}_k^{(i)}), \text{concat}(\boldsymbol{P}_v^{(i)}, \boldsymbol{C}\boldsymbol{W}_v^{(i)})), \tag{5}$$

$\boldsymbol{P}_k$ and $\boldsymbol{P}_v$ are split into $N_h$ head vectors respectively and $\boldsymbol{P}_k^{(i)}, \boldsymbol{P}_v^{(i)} \in \mathbb{R}^{l \times d/N_h}$ denote the $i$-th head vector. Prompt-tuning (Lester et al., 2021) simplifies prefix-tuning by only prepending to the input word embeddings in the first layer; similar work also includes P-tuning (Liu et al., 2021b).

**LoRA (Hu et al., 2021):** LoRA injects trainable low-rank matrices into transformer layers to approximate the weight updates. For a pre-trained weight matrix $\boldsymbol{W} \in \mathbb{R}^{d \times k}$, LoRA represents its update with a low-rank decomposition $\boldsymbol{W} + \Delta W = \boldsymbol{W} + \boldsymbol{W}_\text{down}\boldsymbol{W}_\text{up}$, where $\boldsymbol{W}_\text{down} \in \mathbb{R}^{d \times r}, \boldsymbol{W}_\text{up} \in \mathbb{R}^{r \times k}$ are tunable parameters. LoRA applies this update to the query and value projection matrices $(\boldsymbol{W}_q, \boldsymbol{W}_v)$ in the multi-head attention sub-layer, as shown in Figure 1. For a specific input $\boldsymbol{x}$ to the linear projection in multi-head attention, LoRA modifies the projection output $\boldsymbol{h}$ as:

$$\boldsymbol{h} \leftarrow \boldsymbol{h} + s \cdot \boldsymbol{x}\boldsymbol{W}_\text{down}\boldsymbol{W}_\text{up}, \tag{6}$$

---

[2]In an encoder-decoder architecture, the transformer decoder usually has another multi-head cross-attention module between the self-attention and FFN, which we omit here for simplicity.

[3]Below, we sometimes ignore the head index $i$ to simplify notation when there is no confusion.

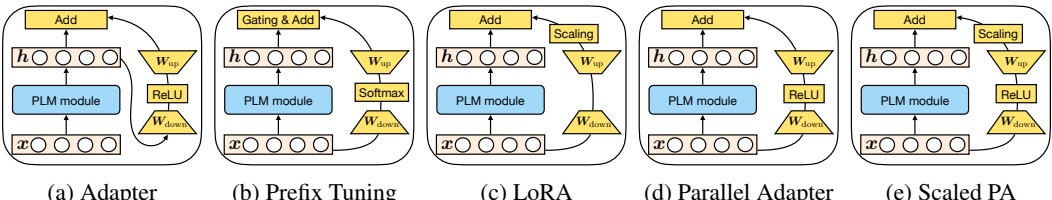

(a) Adapter     (b) Prefix Tuning     (c) LoRA     (d) Parallel Adapter     (e) Scaled PA

Figure 3: Graphical illustration of existing methods and the proposed variants. "PLM module" represents a certain sublayer of the PLM (e.g. attention or FFN) that is frozen. "Scaled PA" denotes scaled parallel adapter. We do not include multi-head parallel adapter here to save space.

where $s \geq 1$ is a tunable scalar hyperparameter.[4]

**Others:** Other parameter-efficient tuning methods include BitFit (Ben Zaken et al., 2021), which only fine-tunes bias vectors in the pre-trained model, and diff-pruning (Guo et al., 2021), which learns a sparse parameter update vector.

## 3 BRIDGING THE GAP – A UNIFIED VIEW

We first derive an equivalent form of prefix tuning to establish its connection with adapters. We then propose a unified framework for parameter-efficient tuning that includes several state-of-the-art methods as instantiations.

### 3.1 A CLOSER LOOK AT PREFIX TUNING

Eq. 5 describes the mechanism of prefix tuning which changes the attention module through prepending $l$ learnable vectors to the original attention keys and values. Here, we derive an equivalent form of Eq. 5 and provide an alternative view of prefix tuning:[5]

$$
\begin{aligned}
\text{head} &= \text{Attn}(\boldsymbol{x}\boldsymbol{W}_q, \text{concat}(\boldsymbol{P}_k, \boldsymbol{C}\boldsymbol{W}_k), \text{concat}(\boldsymbol{P}_v, \boldsymbol{C}\boldsymbol{W}_v)) \\
&= \text{softmax}\left(\boldsymbol{x}\boldsymbol{W}_q\text{concat}(\boldsymbol{P}_k, \boldsymbol{C}\boldsymbol{W}_k)^\top\right) \begin{bmatrix} \boldsymbol{P}_v \\ \boldsymbol{C}\boldsymbol{W}_v \end{bmatrix} \\
&= (1 - \lambda(\boldsymbol{x}))\text{softmax}(\boldsymbol{x}\boldsymbol{W}_q\boldsymbol{W}_k^\top \boldsymbol{C}^\top)\boldsymbol{C}\boldsymbol{W}_v + \lambda(\boldsymbol{x})\text{softmax}(x\boldsymbol{W}_q\boldsymbol{P}_k^\top)\boldsymbol{P}_v \\
&= (1 - \lambda(\boldsymbol{x}))\underbrace{\text{Attn}(\boldsymbol{x}\boldsymbol{W}_q, \boldsymbol{C}\boldsymbol{W}_k, \boldsymbol{C}\boldsymbol{W}_v)}_{\text{standard attention}} + \lambda(\boldsymbol{x})\underbrace{\text{Attn}(\boldsymbol{x}\boldsymbol{W}_q, \boldsymbol{P}_k, \boldsymbol{P}_v)}_{\text{independent of } \boldsymbol{C}},
\end{aligned}
\tag{7}
$$

where $\lambda(\boldsymbol{x})$ is a scalar that represents the sum of normalized attention weights on the prefixes:

$$
\lambda(\boldsymbol{x}) = \frac{\sum_i \exp(\boldsymbol{x}\boldsymbol{W}_q\boldsymbol{P}_k^\top)_i}{\sum_i \exp(\boldsymbol{x}\boldsymbol{W}_q\boldsymbol{P}_k^\top)_i + \sum_j \exp(\boldsymbol{x}\boldsymbol{W}_q\boldsymbol{W}_k^\top \boldsymbol{C}^\top)_j}.
\tag{8}
$$

Note that the first term in Eq. 7, $\text{Attn}(\boldsymbol{x}\boldsymbol{W}_q, \boldsymbol{C}\boldsymbol{W}_k, \boldsymbol{C}\boldsymbol{W}_v)$, is the original attention without prefixes, whereas the second term is a position-wise modification independent of $\boldsymbol{C}$. Eq. 7 gives an alternative view of prefix tuning that essentially applies a position-wise modification to the original head attention output $\boldsymbol{h}$ through linear interpolation:

$$
\boldsymbol{h} \leftarrow (1 - \lambda(\boldsymbol{x}))\boldsymbol{h} + \lambda(\boldsymbol{x})\Delta\boldsymbol{h}, \quad \Delta\boldsymbol{h} := \text{softmax}(\boldsymbol{x}\boldsymbol{W}_q\boldsymbol{P}_k^\top)\boldsymbol{P}_v.
\tag{9}
$$

**The Connection with Adapters:** We define $\boldsymbol{W}_1 = \boldsymbol{W}_q\boldsymbol{P}_k^\top$, $\boldsymbol{W}_2 = \boldsymbol{P}_v$, $f = \text{softmax}$, and rewrite Eq. 9:

$$
\boldsymbol{h} \leftarrow (1 - \lambda(\boldsymbol{x}))\boldsymbol{h} + \lambda(\boldsymbol{x})f(\boldsymbol{x}\boldsymbol{W}_1)\boldsymbol{W}_2,
\tag{10}
$$

which reaches a very similar form to the adapter function in Eq. 4, except that prefix tuning is performing weighted addition while the adapter one is unweighted.[6] Figure 3b demonstrates the

---

[4]The public code of LoRA at https://github.com/microsoft/LoRA uses different $s$ in different datasets, and we have verified the value of $s$ could have a significant effect on the results.

[5]Without loss of generalization, we ignore the softmax scaling factor $\sqrt{d}$ for ease of notation.

[6]$\boldsymbol{h}$ in adapters and prefix tuning are usually different, as described more below. However, here we mainly discuss the functional form as adapters can, in principle, be inserted at any position.

Table 1: Parameter-efficient tuning methods decomposed along the defined design dimensions. Here, for clarity, we directly write the adapter nonlinear function as ReLU which is commonly used. The bottom part of the table exemplifies new variants by transferring design choices of existing approaches.

| Method | $\Delta h$ functional form | insertion form | modified representation | composition function |
|---|---|---|---|---|
| | **Existing Methods** | | | |
| Prefix Tuning | $\text{softmax}(x W_q P_k^\top) P_v$ | parallel | head attn | $h \leftarrow (1 - \lambda)h + \lambda \Delta h$ |
| Adapter | $\text{ReLU}(h W_{\text{down}}) W_{\text{up}}$ | sequential | ffn/attn | $h \leftarrow h + \Delta h$ |
| LoRA | $x W_{\text{down}} W_{\text{up}}$ | parallel | attn key/val | $h \leftarrow h + s \cdot \Delta h$ |
| | **Proposed Variants** | | | |
| Parallel adapter | $\text{ReLU}(h W_{\text{down}}) W_{\text{up}}$ | parallel | ffn/attn | $h \leftarrow h + \Delta h$ |
| Muti-head parallel adapter | $\text{ReLU}(h W_{\text{down}}) W_{\text{up}}$ | parallel | head attn | $h \leftarrow h + \Delta h$ |
| Scaled parallel adapter | $\text{ReLU}(h W_{\text{down}}) W_{\text{up}}$ | parallel | ffn/attn | $h \leftarrow h + s \cdot \Delta h$ |

computation graph of prefix tuning from this view, which allows for abstraction of prefix tuning as a plug-in module like adapters. Further, we note that $W_1 \in \mathbb{R}^{d_h \times l}$ and $W_2 \in \mathbb{R}^{l \times d_h}$ are low-rank matrices when $l$ is small, and thus they function similarly to the $W_{\text{down}}$ and $W_{\text{up}}$ matrices in adapters. This view also suggests that the number of prefix vectors, $l$, plays a similar role to the bottleneck dimension $r$ in adapters: they both represent the rank limitation of computing the modification vector $\Delta h$. Thus we also refer $l$ as the bottleneck dimension. Intuitively, the rank limitation implies that $\Delta h$ is a linear combination of *the same* $l$ (or $\leq l$) basis vectors for any $x$.

**The Difference from Adapters:** In addition to the gating variable $\lambda$, we emphasize three differences between prefix tuning and adapters. (1) As demonstrated in Figure 3, prefix tuning uses $x$, the input of the PLM layer, to compute $\Delta h$, while adapters use $h$, the output of the PLM layer. Thus, prefix tuning can be thought of as a "parallel" computation to the PLM layer, whereas the typical adapter is "sequential" computation. (2) Adapters are more flexible with respect to where they are inserted than prefix tuning: adapters typically modify attention or FFN outputs, while prefix tuning only modifies the attention output of each head. Empirically, this makes a large difference as we will show in §4.4. (3) Eq. 10 applies to each attention head, while adapters are always single-headed, which makes prefix tuning more expressive: head attention is of dimension $d/N_h$ – basically we have full rank updates to each attention head if $l \geq d/N_h$, but we only get full-rank updates to the whole attention output with adapters if $r \geq d$. Notably, prefix tuning is not adding more parameters than adapters when $l = r$.[7] We empirically validate such multi-head influence in §4.4.

## 3.2 THE UNIFIED FRAMEWORK

Inspired by the connections between prefix tuning and adapters, we propose a general framework that aims to unify several state-of-the-art parameter-efficient tuning methods. Specifically, we cast them as learning a modification vector $\Delta h$, which is applied to various hidden representations. Formally, we denote the hidden representation to be directly modified as $h$, and the direct input to the PLM sub-module that computes $h$ as $x$ (e.g. $h$ and $x$ can be the attention output and input respectively). To characterize this modification process, we define a set of design dimensions, and different methods can be instantiated by varying values along these dimensions. We detail the design dimensions below, and illustrate how adapters, prefix tuning, and LoRA fall along them in Table 1:

**Functional Form** is the specific function that computes $\Delta h$. We have detailed the functional form for adapters, prefix tuning, and LoRA in Eq. 4, 6, and 10 respectively. The functional forms of all these methods are similar with a `proj_down` $\rightarrow$ `nonlinear` $\rightarrow$ `proj_up` architecture, while "nonlinear" degenerates to the identity function in LoRA.

**Modified Representation** indicates which hidden representation is directly modified.[8]

**Insertion Form** is how the added module is inserted into the network. As mentioned in the previous section and shown in Figure 3, traditionally adapters are inserted at a position in a sequential manner, where both the input and output are $h$. Prefix tuning and LoRA – although not originally described in this way – turn out to be equivalent to a parallel insertion where $x$ is the input.

---

[7]We will detail in §4.1 the number of parameters added of different methods.

[8]Strictly speaking, all the hidden representations would be indirectly influenced by modifying the ones before them. Here we refer to the position being *directly* modified by the added module.

**Composition Function** is how the modified vector $\Delta h$ is composed with the original hidden representation $h$ to form the new hidden representation. For example, adapters perform simple additive composition, prefix tuning uses a gated additive composition as shown in Eq. 10, and LoRA scales $\Delta h$ by a constant factor and adds it to the original hidden representation as in Eq. 6.

We note that many other methods not present in Table 1 fit into this framework as well. For example, prompt tuning modifies the head attention in the first layer in a way similar to prefix tuning, and various adapter variants (Pfeiffer et al., 2021; Mahabadi et al., 2021) can be represented in a similar way as adapters. Critically, the unified framework allows us to study parameter-efficient tuning methods along these design dimensions, identify the critical design choices, and potentially transfer design elements across approaches, as in the following section.

### 3.3 TRANSFERRING DESIGN ELEMENTS

Here, and in Figure 3, we describe just a few novel methods that can be derived through our unified view above by transferring design elements across methods: (1) *Parallel Adapter* is the variant by transferring the parallel insertion of prefix tuning into adapters. Interestingly, while we motivate the parallel adapter due to its similarity to prefix tuning, concurrent work (Zhu et al., 2021) independently proposed this variant and studied it empirically; (2) *Multi-head Parallel Adapter* is a further step to make adapters more similar to prefix tuning: we apply parallel adapters to modify head attention outputs as prefix tuning. This way the variant improves the capacity for free by utilizing the multi-head projections as we discuss in §3.1. (3) *Scaled Parallel Adapter* is the variant by transferring the composition and insertion form of LoRA into adapters, as shown in Figure 3e.

Our discussion and formulation so far raise a few questions: Do methods varying the design elements above exhibit distinct properties? Which design dimensions are particularly important? Do the novel methods described above yield better performance? We answer these questions next.

## 4 EXPERIMENTS

### 4.1 GENERAL SETUP

**Datasets:** We study four downstream tasks: (1) XSum (Narayan et al., 2018) is an English summarization dataset where models predict a summary given a news article; (2) English to Romanian translation using the WMT 2016 en-ro dataset (Bojar et al., 2016); (3) MNLI (Williams et al., 2018) is an English natural language inference dataset where models predict whether one sentence entails, contradicts, or is neutral to another. (4) SST2 (Socher et al., 2013) is an English sentiment classification benchmark where models predict whether a sentence's sentiment is positive or negative.

**Setup:** We use BART$_{\text{LARGE}}$ (Lewis et al., 2020) and a multilingual version of it, mBART$_{\text{LARGE}}$ (Liu et al., 2020a), as the underlying pretrained models for XSum and en-ro translation respectively, and we use RoBERTa$_{\text{BASE}}$ (Liu et al., 2019) for MNLI and SST2. We vary the bottleneck dimension within $\{1, 30, 200, 512, 1024\}$ if needed.[9] We mainly study adapters, prefix tuning (prefix), and LoRA which greatly outperform bitfit and prompt tuning in our experiments. In the analysis sections (§4.3-4.5) we insert adapters *either* at the attention or FFN layers for easier analysis, but include the results of inserting at both places in the final comparison (§4.6). We re-implement these methods based on their respective public code.[10] We use the huggingface transformers library (Wolf et al., 2020) for our implementation. Complete setup details can be found in Appendix A.

**Evaluation:** We report ROUGE 1/2/L scores (R-1/2/L, Lin (2004)) on the XSum test set, BLEU scores (Papineni et al., 2002) on the en-ro test set, and accuracy on the MNLI and SST2 dev set. For MNLI and SST2, we take the median of five random runs. We also report the number of tuned parameters relative to that in full fine-tuning (#params).

**Number of Tunable Parameters:** BART and mBART have an encoder-decoder structure that has three types of attention: encoder self-attention, decoder self-attention, and decoder cross-attention. RoBERTa only has encoder self-attention. For each attention sub-layer, the number of parameters

---

[9]In some settings we use other values to match the number of added parameters of different methods.

[10]We verify that our re-implementation can reproduce adapter and prefix tuning on XSum, and LoRA on MNLI, by comparing with the results of running the original released code.

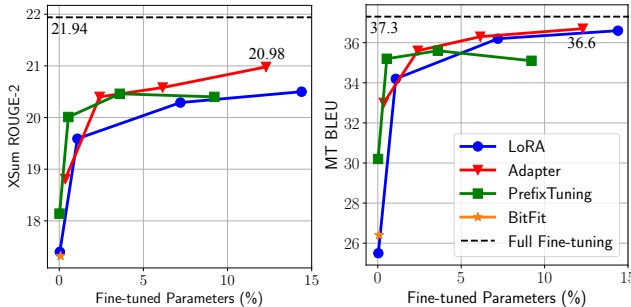

Figure 4: Performance of previous state-of-the-art parameter-efficient tuning methods on XSum (left) and en-ro (right).

Table 2: Accuracy on the dev set of MNLI and SST2. MAM Adapter is proposed in §4.6. Bitfit numbers are from Ben Zaken et al. (2021).

| Method (# params) | MNLI | SST2 |
|---|---|---|
| Full-FT (100%) | $87.6_{\pm.4}$ | $94.6_{\pm.4}$ |
| Bitfit (0.1 %) | 84.7 | 93.7 |
| Prefix (0.5%) | $86.3_{\pm.4}$ | $94.0_{\pm.1}$ |
| LoRA (0.5%) | $87.2_{\pm.4}$ | $94.2_{\pm.2}$ |
| Adapter (0.5%) | $87.2_{\pm.2}$ | $94.2_{\pm.1}$ |
| MAM Adapter (0.5%) | $\mathbf{87.4}_{\pm.3}$ | $94.2_{\pm.3}$ |

Table 3: Comparison of different insertion forms for adapters, i.e. sequential adapter (SA) and parallel adapter (PA). We include the results of prefix tuning as a reference point.

| Method | # params | XSum (R-1/2/L) | MT (BLEU) |
|---|---|---|---|
| Prefix, $l$=200 | 3.6% | 43.40/20.46/35.51 | 35.6 |
| SA (attn), $r$=200 | 3.6% | 42.01/19.30/34.40 | 35.3 |
| SA (ffn), $r$=200 | 2.4% | 43.21/19.98/35.08 | 35.6 |
| PA (attn), $r$=200 | 3.6% | 43.58/20.31/35.34 | 35.6 |
| PA (ffn), $r$=200 | 2.4% | **43.93/20.66/35.63** | **36.4** |

Table 4: Results on en-ro dataset.

| Method | # params | MT (BLEU) |
|---|---|---|
| PA (attn), $r$=200 | 3.6% | 35.6 |
| Prefix, $l$=200 | 3.6% | 35.6 |
| MH PA (attn), $r$=200 | 3.6% | 35.8 |
| Prefix, $l$=30 | 0.1% | 35.2 |
| -gating, $l$=30 | 0.1% | 34.9 |
| PA (ffn), $r$=30 | 0.1% | 33.0 |
| PA (attn), $r$=30 | 0.1% | 33.7 |
| MH PA (attn), $r$=30 | 0.1% | **35.3** |

used of each method is: (1) prefix tuning prepends $l$ vectors to the keys and values and uses $2 \times l \times d$ parameters; (2) adapter has $\boldsymbol{W}_{\text{down}}$ and $\boldsymbol{W}_{\text{up}}$ thus uses $2 \times r \times d$ parameters; (3) LoRA employs a pair of $\boldsymbol{W}_{\text{down}}$ and $\boldsymbol{W}_{\text{up}}$ for query and value projections, hence uses $4 \times r \times d$ parameters. For the adapter modification at ffn, it uses $2 \times r \times d$ parameters which is the same as adapter at attention. Therefore, for a specific value of $r$ or $l$, prefix tuning uses the same number of parameters as adapters, while LoRA uses more parameters. More details can be found in Appendix B.

## 4.2 THE RESULTS OF EXISTING METHODS

We first overview the results of existing methods on the four tasks. As shown in Figure 4 and Table 2, while existing methods can achieve competitive performance on MNLI and SST2 by tuning fewer than 1% parameters, a large gap is still present if we add 5% parameters in XSum and en-ro. The gap remains significant even though we increase the relative parameter size to >10%. Even larger gaps have been observed in Raffel et al. (2020) on high-resource MT tasks. This shows that many methods that claimed comparable results to full fine-tuning on the GLUE benchmark with an encoder-only model (Guo et al., 2021; Ben Zaken et al., 2021; Mahabadi et al., 2021), or on relatively simple generation benchmarks such as E2E (Novikova et al., 2017) with an encoder-decoder model (Li & Liang, 2021), may not generalize well to other standard benchmarks. The influencing factors could be complicated including the number of training samples, task complexity, or model architecture. We thus advocate for future research on this line to report results on more diverse benchmarks to exhibit a more complete picture of their performance profile. Below, our analysis will mainly focus on the XSum and en-ro datasets to better distinguish different design choices. We note that these two benchmarks are relatively high-resource performed with an encoder-decoder model (BART), while we will discuss the results on MNLI and SST2 with an encoder-only model (RoBERTa) in §4.6.

## 4.3 WHICH INSERTION FORM – SEQUENTIAL OR PARALLEL?

We first study the insertion form design dimension, comparing the proposed parallel adapter (PA) variant to the conventional sequential adapter (SA) over both the attention (att) and FFN modification. We also include prefix tuning as a reference point. As shown in Table 3, prefix tuning, which uses parallel insertion, outperforms attention sequential adapters. Further, the parallel adapter is able to beat sequential adapters in all cases,[11] with PA (ffn) outperforming SA (ffn) by 1.7 R-2 points on

---

[11]More results with different $r$ can be found in Appendix C, which exhibits similar observations.

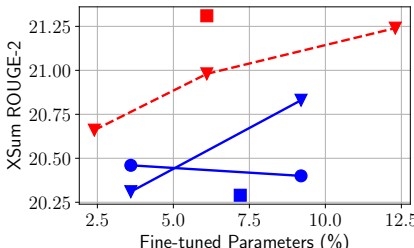 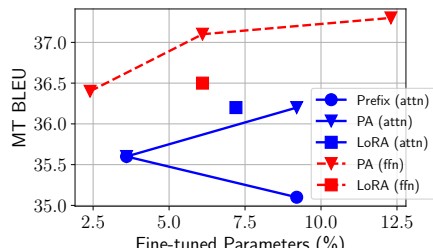

Figure 5: Results on XSum (left) and en-ro (right). PA represents parallel adapter. Blue and red markers apply modifications at attention and FFN sub-layers respectively (best viewed in color).

XSum and 0.8 BLEU points on en-ro respectively. Given the superior results of parallel adapters over sequential adapters, we focus on parallel adapter results in following sections.

### 4.4 WHICH MODIFIED REPRESENTATION – ATTENTION OR FFN?

**Setup:** We now study the effect of modifying different representations. We mainly compare attention and FFN modification. For easier analysis we categorize methods that modifies any hidden representations in the attention sub-layer (e.g. the head output, query, etc) as modifying the attention module. We compare parallel adapters at attention and FFN and prefix tuning. We also transfer the FFN modification to LoRA to have a LoRA (ffn) variant for a complete comparison. Specifically, we use LoRA to approximate the parameter updates for the FFN weights $W_1 \in \mathbb{R}^{d \times d_m}$ and $W_2 \in \mathbb{R}^{d_m \times d}$. In this case $W_{up}$ in LoRA for $W_1$ (similar for $W_{down}$ of $W_2$) would have dimensions of $r \times d_m$, where $d_m = 4d$ as described in §2.1. Thus we typically use smaller $r$ for LoRA (ffn) than other methods to match their overall parameter size in later experiments.

**Results:** As shown in Figure 5, *any* method with FFN modification outperforms *all* the methods with attention modification in all cases (the red markers are generally above all the blue ones, the only exception is ffn-PA with 2.4% params), often with fewer parameters. Second, the same method applied at FFN always improves over its attention counterpart. For example, LoRA (ffn) improves LoRA (attn) by 1 R-2 points on XSum. We also highlight that prefix tuning does not keep improving when we further increase the capacity, which is also observed in Li & Liang (2021). These results suggest that *FFN modification can utilize the added parameters more effectively than attention, no matter what the functional form or composition function is.* We hypothesize that this is because the FFN learns task-specific textual patterns (Geva et al., 2021), while attention learns pairwise positional interactions which do not require large capacity for adapting to new tasks.

**Is the story different when we use** $0.1\%$ **parameters?** In §3.1 we reason that prefix tuning is more expressive than adapters (attn), which, however, is not reflected in Figure 5. We conjecture that this is because multi-head attention is only superior when the parameter budget is small. To validate this hypothesis, we compare prefix tuning to parallel adapters when they add $0.1\%$ of the pretrained parameters. To ablate the impact of the composition function, we also report the results of removing the gating in prefix tuning as $h + \Delta h$. We include the results of the multi-head parallel adapter variant (MH PA) described in §3.3. As shown in Table 4, the multi-head methods – prefix tuning and MH PA (attn) – outperform all others by at least 1.6 BLEU points when using 0.1% of the parameters. Surprisingly, reducing $l$ from 200 to 30 only causes 0.4 BLEU loss for prefix tuning while PA (attn) loses 1.9 points. The gating composition function in prefix tuning slightly helps the results by 0.3 points. We highlight that the MH parallel adapter improves the single-headed version by 1.6 points, which again verifies the effectiveness of the multi-head formulation.

Combining the results in Figure 5 and Table 4, we conclude that *modifying head attention shows the best results when the parameter budget is very small, while the FFN can better utilize modifications at larger capacities.* This suggests that it may be effective to allocate a larger parameter budget to FFN modification instead of treating attention and FFN equally as in Houlsby et al. (2019).

### 4.5 WHICH COMPOSITION FUNCTION?

We have presented three composition functions in §3.2: simple addition (adapter), gated addition (prefix tuning) and scaled addition (LoRA). As it is unnatural to incorporate the exact gated addition into methods whose functional form does not use softmax, we examine the other two by

Table 6: Comparison of various parameter-efficient tuning methods and the proposed variants. "†" are results copied from Lewis et al. (2020) and Liu et al. (2020b). We could not reproduce exactly the same full fine-tuning numbers with the same hyperparameters or even searching them. The reason may be the different libraries which the training code is based on – full fine-tuning is very sensitive to training hyperparameters. For the most performant methods we run with 3 random seeds and report mean and standard deviation.

| Method | # params | XSum (R-1/2/L) | MT (BLEU) |
|---|---|---|---|
| Full fine-tuning† | 100% | 45.14/22.27/37.25 | 37.7 |
| Full fine-tuning (our run) | 100% | 44.81/21.94/36.83 | 37.3 |
| Bitfit (Ben Zaken et al., 2021) | 0.1% | 40.64/17.32/32.19 | 26.4 |
| Prompt tuning (Lester et al., 2021) | 0.1% | 38.91/15.98/30.83 | 21.0 |
| Prefix tuning (Li & Liang, 2021), $l$=200 | 3.6% | 43.40/20.46/35.51 | 35.6 |
| Pfeiffer adapter (Pfeiffer et al., 2021), $r$=600 | 7.2% | 44.03/20.89/35.89$_{\pm.13/.10/.08}$ | 36.9$_{\pm.1}$ |
| LoRA (ffn), $r$=102 | 7.2% | 44.53/21.29/36.28$_{\pm.14/.07/.10}$ | 36.8$_{\pm.3}$ |
| Parallel adapter (PA, ffn), $r$=1024 | 12.3% | 44.71/21.41/36.41$_{\pm.16/.17/.16}$ | 37.2$_{\pm.1}$ |
| PA (attn, $r$=30) + PA (ffn, $r$=512) | 6.7% | 44.29/21.06/36.12$_{\pm.31/.19/.18}$ | 37.2$_{\pm.1}$ |
| Prefix tuning (attn, $l$=30) + LoRA (ffn, $r$=102) | 6.7% | 44.84/21.71/36.77$_{\pm.07/.05/.03}$ | 37.0$_{\pm.1}$ |
| MAM Adapter (our variant, $l$=30, $r$=512) | 6.7% | **45.06/21.90/36.87**$_{\pm.08/.01/.04}$ | **37.5**$_{\pm.1}$ |

ablating on LoRA and comparing with the proposed scaled parallel adapter (Scaled PA), we constrain modified representation to be FFN since it is generally more effective as shown in §4.4.

Table 5 reports the results on XSum. We set $r$ as 512 for adapters and 102 for LoRA so that their tuned parameter sizes are the same. We select $s$ based on the R-2 score on the dev set. We observe that LoRA ($s = 4$) performs better than parallel adapter. However, the advantage disappears if we remove the scaling by setting $s = 1$. Through plugging the composition function of LoRA into parallel adapter, the resulted Scaled PA improves the vanilla parallel adapter by 0.56 ROUGE-2 points. We also experiment with a learned scalar which does not give better results. Therefore, we conclude that *the scaling composition function is better than the vanilla additive one while being easily applicable.*

Table 5: Results on XSum when using different composition functions. The modified representation is FFN. The bottleneck dimension $r = 512$ for (Scaled) PA and $r = 102$ for LoRA.

| Method (# params) | XSum (R-1/2/LSum) |
|---|---|
| LoRA (6.1%), $s$=4 | 44.59/21.31/36.25 |
| LoRA (6.1%), $s$=1 | 44.17/20.83/35.74 |
| PA (6.1%) | 44.35/20.98/35.98 |
| Scaled PA (6.1%), $s$=4 | **44.85/21.54/36.58** |
| Scaled PA (6.1%), trainable $s$ | 44.56/21.31/36.29 |

### 4.6 AN EFFECTIVE INTEGRATION BY TRANSFERRING FAVORABLE DESIGN ELEMENTS

We first highlight three findings in previous sections: (1) Scaled parallel adapter is the best variant to modify FFN; (2) FFN can better utilize modification at larger capacities; and (3) modifying head attentions like prefix tuning can achieve strong performance with only 0.1% parameters. Inspired by them, we mix and match the favorable designs behind these findings: specifically, we use prefix tuning with a small bottleneck dimension ($l = 30$) at the attention sub-layers and allocate more parameter budgets to modify FFN representation using the scaled parallel adapter ($r = 512$). Since prefix tuning can be viewed as a form of adapter in our unified framework, we name this variant as *Mix-And-Match adapter (MAM Adapter)*. In Table 6, we compare MAM adapter with various parameter-efficient tuning methods. For completeness, we also present results of other combination versions in Table 6: using parallel adapters at both attention and FFN layers and combining prefix tuning (attn) with LoRA (ffn) – both of these combined versions can improve over their respective prototypes. However, MAM Adapter achieves the best performance on both tasks and is able to match the results of our full fine-tuning by only updating 6.7% of the pre-trained parameters. In Table 2, we present the results of MAM Adapter on MNLI and SST2 as well, where MAM Adapter achieves comparable results to full fine-tuning by adding only 0.5% of pretrained parameters.

## 5 DISCUSSION

We provide a unified framework for several performant parameter-tuning methods, which enables us to instantiate a more effective model that matches the performance of full fine-tuning method through transferring techniques across approaches. We hope our work can provide insights and guidance for future research on parameter-efficient tuning.

ETHICS STATEMENT

Our work proposes a method for efficient fine-tuning of pre-trained models, in particular language models. Pre-trained language models have a wide variety of positive applications, such as the applications to summarization, translation, or language understanding described in our paper. At the same time, there are a number of ethical concerns with language models in general, including concerns regarding the generation of biased or discriminative text (Bordia & Bowman, 2019), the leakage of private information from training data (Carlini et al., 2020), and environmental impact of training or tuning them (Strubell et al., 2019).

Our method attempts to train language models making minimal changes to their pre-existing parameters. While it is an interesting research question whether parameter-efficient fine-tuning methods exacerbate, mitigate, or make little change to issues such as bias or information leakage, to our knowledge no previous work has examined this topic. It is an interesting avenue for future work.

With respect to environmental impact, the methods proposed in this paper add a small number of extra parameters and components to existing models, and thus they have a nominal negative impact on training and inference time – for example, the final MAM Adapter needs 100% - 150% training time of full fine-tuning in our four benchmarks since parameter-efficient tuning typically needs more epochs to converge; the inference time is roughly the same as the model obtained by full fine-tuning. On the other hand, as the methods proposed in this paper may obviate the need for full fine-tuning, this may also significantly reduce the cost (in terms of memory/deployed servers) of serving models. Notably, the great majority of the experimentation done for this paper was performed on a data center powered entirely by renewable energy.

REPRODUCIBILITY STATEMENT

In addition to the setup description in §4.1, we have detailed the complete experiments setup such as batch size, optimizer, learning rates in Appendix A. Besides, we have publicized our source code. These resources should be sufficient to reproduce results of the paper.

ACKNOWLEDGEMENT

We thank the anonymous reviewers for their comments. This work was supported in part by the CMU-Portugal MAIA Project, a Baidu PhD Fellowship for Junxian He, and a CMU Presidential Fellowship for Chunting Zhou.

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

# A  EXPERIMENTS

## A.1  SETUPS

Table 7: Dataset Statistics of the four tasks.

| Dataset | #train | #dev | #test |
|---------|--------|------|-------|
| XSum | 204,045 | 113,332 | 113,334 |
| WMT16 en-ro | 610,320 | 1,999 | 1,999 |
| MNLI | 392,702 | 9815 | 9832 |
| SST-2 | 67,349 | 872 | 1,821 |

We implement all the parameter-efficient tuning methods using the huggingface transformers library (Wolf et al., 2020). We use BART$_{\text{LARGE}}$(Lewis et al., 2020) and mBART$_{\text{LARGE}}$ (Liu et al., 2020b) (mBART-cc25) for the summarization and machine translation tasks respectively, and we use RoBERTa$_{\text{BASE}}$ (Liu et al., 2019) for MNLI and SST2. BART$_{\text{LARGE}}$ and mBART$_{\text{LARGE}}$ have the same encoder-decoder architectures. mBART$_{\text{LARGE}}$ is pre-trained on 25 languages. We use their public checkpoints from the transformers library in experiments. For MT and classifications tasks, the max token lengths of training data are set to be 150 and 512 respectively. For XSum, we set the max length of source articles to be 512 and the max length of the target summary to be 128. The detailed dataset statistics is present in Table 7. In our summarization experiments, we only use 1600 examples for validation to save time.

While we vary the bottleneck dimension within $\{1, 30, 512, 1024\}$ as mentioned in §4.1, we test bottleneck dimension 1024 only when the modified representation is FFN, because the training of prefix tuning does not fit into 48GB GPU memory when $l = 1024$. While other methods do not have memory issues, we keep the bottleneck dimension of attention modification at most 512 to have a relatively fair comparison with prefix tuning. For LoRA we always tune its scaling hyperparameters $s$ on the dev set.

## A.2  TRAINING AND EVALUATION

We present some training hyperparameters of parameter-efficient tuning methods in Table 8. For all the tasks, we train with the Adam optimizer (Kingma & Ba, 2015), and use a polynomial learning rate scheduler that linearly decays the learning rate throughout training. We set the warm up steps of learning rate to be 0 for both MT and summarization tasks, and for the classification tasks, learning rate is linearly warmed up from 0 for the first 6% of the total training steps before decay. For full fine-tuning we set these training hyperparameters following Lewis et al. (2020) (XSum), Liu et al. (2020b) (en-ro), and (Liu et al., 2019) (MNLI and SST2). We also did hyperparameter search in the full fine-tuning case to try to reproduce their results. We set dropout rate to be 0.1 for all the tasks. We use ROUGE-2 and perplexity as the validation metrics for summarization and MT respectively.

For MT and text summarization, we use beam search for decoding and set the number of beams to be 6 and 5 following previous work (Li & Liang, 2021; Liu et al., 2020b). The min and max generation lengths for summarization and MT are set to be (10, 60) and (1, 200) respectively.

## A.3  OTHER EXPERIMENTAL DETAILS

**Prefix Tuning:**  Following Li & Liang (2021), we reparameterize the prefix vectors by a MLP network which is composed of a small embedding matrix and a large feedforward neural network. This is conducive for learning due to the shared parameters across all layers.

**LoRA:**  LoRA and adapter employ different parameter initialization methods: LoRA uses a random Kaiming uniform (He et al., 2015) initialization for $W_{\text{down}}$ and zero for $W_{\text{up}}$ (LoRA init), while adapters use the same initialization as BERT (Devlin et al., 2019). We found it beneficial to use the same initialization method as LoRA in scaled PA.

Table 8: Training hyperparameters of parameter-efficient tuning methods on the four tasks. lr and ls represents learning rate and label smoothing respectively.

| Tasks | lr | batch size | ls | max grad norm | weight decay | train steps |
|---|---|---|---|---|---|---|
| XSum | 5e-5 | 64 sents | 0.1 | 0.1 | 0.01 | 100K |
| enro MT | 5e-5 | 16384 tokens | 0.1 | 1.0 | 0.01 | 50K |
| MNLI/SST2 | 1e-4 | 32 sents | 0 | 1.0 | 0.1 | 10 epochs |

## B  COMPUTATION OF TUNABLE PARAMETERS

Table 9: Number of attention or FFN sub-layers in each layer of the pre-trained models.

| | BART/mBART$_{\text{LARGE}}$ | RoBERTa$_{\text{BASE}}$ |
|---|---|---|
| $N_{\text{attn}}$ | 3 | 1 |
| $N_{\text{ffn}}$ | 2 | 1 |

Table 10: Number of parameters used at each sub-layer for different methods.

| | $N_{\text{W}}^{\text{attn}}$ | $N_{\text{W}}^{\text{ffn}}$ |
|---|---|---|
| Prefix Tuning | $2ld$ | – |
| Adapter variants | $2rd$ | $2rd$ |
| LoRA | $2 \times 2rd = 4rd$ | $2 \times (rd + 4dr) = 10rd$ |

We compute the number of tunable parameters based on where the tunable module is inserted into and how it is parameterized. The pretrained-models for summarization or MT have an encoder-decoder structure and each has $L$ layers, whereas RoBERTa$_{\text{BASE}}$ for classification tasks only has $L$ encoder layers. To simplify the computation of tunable parameters, we compute the sum of parameter used in one encoder layer and one decoder layer as the parameter overhead of one single layer of the pre-trained encoder-decoder model. Each layer has $N_{\text{attn}}$ sub-layers and $N_{\text{ffn}}$ sub-layers. For the encoder-decoder models, $N_{\text{attn}} = 3$: the encoder self-attention, the decoder self-attention and the decoder cross-attention. For the classification tasks, RoBERTa$_{\text{BASE}}$ only has the encoder self-attention, thus $N_{\text{attn}} = 1$. We present the number of attention and ffn sub-layers for different pre-trained models in Table 10. For modifications applied at the attention sub-layers, the number of tunable parameters is computed by $|\Theta|_{\text{attn}} = N_{\text{W}}^{\text{attn}} \times N_{\text{attn}} \times L$, where $N_{\text{W}}^{\text{attn}}$ denotes the number of parameters ($\boldsymbol{W}_{\text{down}}$ or $\boldsymbol{W}_{\text{up}}$) used for one attention sub-layer. Similarly, the number of tunable parameters for the FFN sub-layers is computed by $|\Theta|_{\text{ffn}} = N_{\text{W}}^{\text{ffn}} \times N_{\text{ffn}} \times L$. In Table 10, we show the number of parameters for one sub-layer. As we have explained in §4.4, LoRA approximates the update of each weight matrix with a pair of $\boldsymbol{W}_{\text{down}}$ and $\boldsymbol{W}_{\text{up}}$, thus LoRA typically uses more parameters with the same $r$ as other methods. Finally, the total number of tunable parameters for prefix tuning, adapter variants and LoRA is $|\Theta| = |\Theta|_{\text{attn}} + |\Theta|_{\text{ffn}}$ as applicable. Prompt tuning prepends $l$ tunable vectors at the input layer and uses $l \times d$ number of parameters. Using MBART/BART as an example, we present the number of parameters used by several representative methods throughout our paper in Table 11, where adapter variants include sequential adapter, parallel adapter, scaled adapter and multi-head adapter.

Table 11: Number of tunable parameters of various parameter-efficient tuning methods with BART/MBART models ($L = 12$) as an example.

| Method | number of parameters |
|---|---|
| Prompt Tuning | $l \times d$ |
| Prefix Tuning (attn) | $2ld \times 3 \times 12$ |
| Adapter variants (attn) | $2rd \times 3 \times 12$ |
| Adapter variants (ffn) | $2rd \times 2 \times 12$ |
| LoRA (attn) | $4rd \times 3 \times 12$ |
| LoRA (ffn) | $10rd \times 2 \times 12$ |
| MAM Adapter (our proposed model) | $2ld \times 3 \times 12 + 2rd \times 2 \times 12$ |

## C  FULL RESULTS ON DIFFERENT BOTTLENECK DIMENSIONS

Table 12: Performance on the test sets of abstractive summarization (XSum) and WMT EN-RO translation.

| Method | # params (%) | XSum (R-1/2/L) | MT BLEU |
|---|---|---|---|
| Modified Representation: attention | | | |
| Prefix Tuning, $r = 200$ | 3.6 | 43.40/20.46/35.51 | 35.6 |
| Prefix Tuning, $r = 512$ | 9.2 | 43.29/20.40/35.37 | 35.1 |
| LoRA, $r = 200$ | 7.2 | 43.09/20.29/35.37 | 36.2 |
| Sequential Adapter, $r = 200$ | 3.6 | 42.01/19.30/34.40 | 35.3 |
| Sequential Adapter, $r = 512$ | 9.2 | 41.05/18.87/33.71 | 34.7 |
| Parallel Adapter, $r = 200$ | 3.6 | 43.58/20.31/35.34 | 35.6 |
| Parallel Adapter, $r = 512$ | 9.2 | 43.99/20.83/35.77 | 36.2 |
| Modified Representation: FFN | | | |
| LoRA, $r = 102$ | 6.1 | 44.59/21.31/36.25 | 36.5 |
| Sequential Adapter, $r = 200$ | 2.4 | 43.21/19.98/35.08 | 35.6 |
| Sequential Adapter, $r = 512$ | 6.1 | 43.72/20.75/35.64 | 36.3 |
| Sequential Adapter, $r = 1024$ | 12.3 | 43.95/21.00/35.90 | 36.7 |
| Parallel Adapter, $r = 200$ | 2.4 | 43.93/20.66/35.63 | 36.4 |
| Parallel Adapter, $r = 512$ | 6.1 | 44.35/20.98/35.98 | 37.1 |
| Parallel Adapter, $r = 1024$ | 12.3 | 44.53/21.24/36.23 | 37.3 |

