# OpenReview forum: "Towards a Unified View of Parameter-Efficient Transfer Learning"
_ICLR.cc/2022/Conference — ICLR 2022 Spotlight_

### Official Review · Reviewer_eByL · 2021-10-26

**Correctness:** 3
**Technical Novelty And Significance:** 3
**Empirical Novelty And Significance:** 3
**Recommendation:** 8
**Confidence:** 4

**Main Review:**

Pros:

- The common framework for different approaches is useful overall. In particular, I liked the section clarifying the connection between Prefix Tuning and Adapters. Although there is nothing groundbreaking here, I believe these frameworks can be useful and help researchers.
- The authors show how the framework can be used to derive new approaches. They also show that these approaches can be more effective than existing ones.
- The experimental setup is well-described overall, with hyperparameters being provided
- This is a hard analysis to perform, with many possible things to change. I feel like the experiment set chosen is convincing overall, with the caveat mentioned below.

Cons:

- Significance of results:

There are no standard deviations for any results in the paper, which makes it hard to assess significance of many results, esp. since some of the performance gaps are small.
For instance, in table 6, authors have a 0.4 BLEU discrepancy in their replication of full fine-tuning performance but draw conclusions on MAM being best based on a 0.2 BLEU gap. The gap is bigger when comparing MAM to methods not introduced by the authors, but still.
The story makes sense but I am not 100% confident in the robustness of the results. Using two decimal numbers for tasks (e.g: XSum) also gives a false impression of precision.
I understand fine-tuning on some of these tasks (MT / XSUM) can be resource-intensive but having standard deviations in even a subset of the experiments would be useful.

- Interpretation of Figure 4 in Section 5.2 and effectiveness of MAM Adapters for encoder models

The authors highlight that while existing methods perform well on MNLI/SST, they are underwhelming on en-ro / XSum. They conclude that this means existing parameter-efficient transfer learning (PETL) approaches are not great for higher-resource / more challenging tasks.
	However, I would point out that this data can also support a different conclusion: existing PETL approaches work well for encoder-only models but are not great for encoder-decoder models. Including the T5 datapoint is also not very relevant since in that case Superglue is treated as a single task (w/ one adapter). Answering “For which tasks / architectures do PETL methods perform well” is an interesting question in itself and I feel like this section expedites this. It is likely that both the architecture and #datapoints in the task matter.
	The author’s choice at the end of section 4.2 is to focus on XSum / en-ro MT. It would be great to highlight that not only are those higher-resource tasks, but they are also generative ones, and thus all the conclusions of this paper might not apply widely to encoder models. Indeed, the results of MAM adapters in table 2 are quite mixed. Right now the paper is claiming more generality than deserved.
	Two possible fixes here: (1) Make claims less general and specific to enc-dec models, (2) Conduct more experiments on encoder-only models

- Writing:
  - The writing is subpar right now. Example: section 4.4: “its counterpart at attention” -> “its attention counterpart”,  “FFN can better utilize modification at larger capacities” -> modifications or “modification of the FFN is better at larger capacities”, etc.
  - Minor: The citation style is often wrong for the sentence. Use \citet more often.
  - Minor: Replace “For classifications” with “for classification tasks” in the Appendix (twice)
  - Typos: Section 5 name Discussions -> Discussion ;  Appendix A3 Learning -> Learning.

- Minor:
  - For Figure 2, I feel like the full fine-tuning number should come from the original baseline, not the replication.
  - The protocol in 4.5 for choosing the scaling parameter for Scaled PA is a bit surprising. I would just suggest that it should be an hyperparameter, the current description makes it seem like it is based on hyperparam tuning for LoRA.
  - It seems to me that 4.5 is about composition function *and* \delta h functional form since LoRA changes both.


~~~~~~~~

Revised score upwards after response, see below


**Summary Of The Paper:**

This paper formulates different approaches for parameter-efficient transfer learning (such as adapters, prefix-tuning, LoRA) under a common framework. Approaches vary alongside different design dimensions: functional form, insertion form (sequential or parallel), which representation is directly modified by the approach and the composition function of the main representation with the adaptation one.

The authors describe how existing approaches fit this framework. Then, they show how changing these design choices leads to new approaches, including parallel adapters (PA), multi-head PA and scaled PA.
The authors then compare their newly introduced methods and existing ones empirically. They find that:
1. Parallel insertion is generally better
2. Under a low parameter budget (~0.1% of original) it is preferable to modify the attention of the transformer. Under a higher budget, changing the feed-forward network is better.
3. Their methods generally perform on par or better than existing approaches.

Finally, the authors combine their insights to develop “Mix-and-match” adapters and show that it does well overall.


**Summary Of The Review:**

Compelling framework to unify parameter-efficient transfer learning approaches, with interesting new methods emerging from it. The set of experiments is well-chosen overall, but the lack of standard deviations and the focus on generative tasks / enc-decoder models makes the results less robust and general. The writing would benefit from more work as well.

---

> ### Author Response · Authors · 2021-11-15
> **We have added standard deviation numbers and clarified some statements in the revision**
>
> Thank you for your time and helpful comments! We address your concerns below:
>
> **-Q1: Standard deviation of results**
>
> We have added standard deviation numbers of XSum and en-ro to Table 6 as well as those of MNLI and SST2 to Table 2 in the revision. For XSum and en-ro we have rerun the most performant methods in Table 6 with 3 different random seeds and updated mean/standard deviation numbers in Table 6 – as shown in Table 6, the standard deviation is mostly small for both XSum and en-ro.  For the analysis experiments (Section 4.3-4.5) the performance difference between methods is typically large, thus we do not rerun all of them with different random seeds due to the expensive computation. We hope the added standard deviation numbers would clear the concerns on the significance of the results.
>
> Regarding the “small gap” between some methods in Table 6, we would like to emphasize that, as the reviewer acknowledged, the relatively small gap is present only between the new variants presented in this paper which are part of our contributions. Compared to the best existing method in Table 6 (Pfeiffer adapter), MAM adapter outperforms it by 1 ROUGE-2 point on XSum and 0.6 BLEU point on en-ro on average across 3 random runs, which we believe is a significant improvement.
>
> **-Q2: Using two decimal numbers for tasks (e.g: XSum) also gives a false impression of precision**
>
> Most summarization work report ROUGE scores with two decimal numbers [1,2,3,4,5], thus we followed them to report two decimal numbers as well. We agree that using two decimal numbers may not be well-warranted, and we will consider changing it later while keeping it as it is in the rebuttal period.
>
> **-Q3: Interpretation of Figure 4 in Section 4.2**
>
> You are right. The phenomenon present in Figure 4 and Table 2 may be not only due to #datapoints or task difficulties, but architectures (encoder-only v.s. encoder-decoder) may matter as well. Indeed, it is likely that the reason lies in the combination of these factors and there is not a single factor that could entirely explain the difference. As the reviewer noted, however, “For which tasks/architectures do PETL methods perform well” is an interesting and non-trivial question itself, and answering this is beyond the scope of this paper. We emphasize that our goal of Section 4.2 is not to answer this question, but to point out that existing PETL methods still lag behind full-finetuning on some standard benchmark tasks, even though 10% of the parameters are tuned. This is because we feel that most previous work – by only reporting numbers on the GLUE benchmark – may unintentionally leave an impression that PETL methods can match full-finetuning easily with <1% of full parameters on any task, which is clearly not the case. We have revised the statement regarding this interpretation in Section 4.2 to make it more accurate by explicitly stating that the reasons could be related to architectures as well. Please check out our updates, and thank you for the advice, we feel the claims are more precise now.

---

> > ### Author Response · Authors · 2021-11-15
> > **Cont'd**
> >
> > **-Q4: Experiments focus on generation tasks, and conclusions of this paper might not apply widely to encoder models**
> >
> > This is a good point. Several previous works with different methods have shown comparable results to full fine-tuning on GLUE tasks when tuning only <1% of full parameters, thus we chose not to focus on GLUE since (1) a very small parameter budget (<1%) seems sufficient for different methods to match results of full fine-tuning on GLUE, thus it is difficult to obtain strong empirical differences; and (2) it may not be practically meaningful to further constrain the parameter budget (e.g. <0.01%) to distinguish approaches given that 1% parameters only take ~10MB on disk for RoBERTa-large.
> >
> > We agree with the reviewer’s point on the applicability of some of the results, and below we would like to categorize the main contributions of this paper, and highlight which ones of them are general, which ones are not, and how we revised the paper to reflect the reviewer’s comments:
> >
> > 1. The main technical contributions of this paper are the connections between existing methods and the proposed unified framework (Sections before experiments). This part is agnostic to architecture or tasks.
> >
> > 2. The experimental examination of design dimensions (Section 4.3 - 4.5) is on summarization and MT tasks. We gave valid reasons why we do not perform them on GLUE, but still, the conclusions from this part could be potentially constrained to encoder-decoder models, as mentioned by the reviewer. We agree on this point, thus in the revision we have explicitly highlighted this limitation at the end of Section 4.2. Please check it out and let us know if you have any further concerns.
> >
> > 3. The proposed MAM Adapter performs well, achieving better or comparable results to previous methods. This conclusion is general as reflected by both Table 2 on MNLI/SST2 and Table 6 on XSum/en-ro. The reviewer mentioned that “the results of MAM adapter in table 2 are quite mixed” which we kindly disagree with. In Table 2 the only baseline number that outperformed MAM adapter was the SST2 number from Adapter (0.9%), but we note that it is tuning more parameters than MAM adapter thus the comparison was not fair (we set its bottleneck dimension to be the same as prefix tuning and thus Houlsby adapter [6] used 2x #params compared to others). In the revision, we have replaced it with a fair adapter baseline with 0.5% parameters in Table 2, which shows similar results to the MAM adapter. Therefore, the MAM adapter indeed works well generally, achieving better or comparable results to previous methods across MNLI, SST2, XSum, and en-ro benchmarks.
> >
> > **-Q5: Writing**
> >
> > We have revised the places you mentioned as well as some citation styles. Thanks for the suggestion!
> >
> > **-Q6: protocol in Section 4.5 for choosing scaling parameter**
> >
> > The scaling factor is indeed a hyperparameter for both LoRA and Scaled PA. We select it within {1, 2, 4} depending on the validation performance. The choice of scaling factors in LoRA and Scaled PA are independent, we have revised the description in Section 4.5 to make it more clear.
> >
> > **-Q7: Section 4.5 is about composition function and \delta h functional form**
> >
> > Good point. We can draw conclusions with respect to composition function by comparing LoRA (s=4) to LoRA (s=1), or comparing PA to Scaled PA; we can also argue about \delta h functional form by comparing LoRA (s=1) to PA, or comparing LoRA (s=4) to Scaled PA. In this paper, we intend to focus on the former in Section 4.5, composition function, to simplify the analysis.
> >
> > [1] Rush et al. A Neural Attention Model for Abstractive Sentence Summarization. EMNLP 2015
> > [2] See et al. Get To The Point: Summarization with Pointer-Generator Networks. ACL 2017.
> > [3] Li et al. Prefix-tuning: Optimizing continuous prompts for generation. ACL 2021.
> > [4] Lewis et al. BART: Denoising Sequence-to-Sequence Pre-training for Natural Language Generation, Translation, and Comprehension. ACL 2020.
> > [5] Zhang et al. Pegasus: Pre-training with extracted gap-sentences for abstractive summarization. ICML 2020.
> > [6] Houlsby et al. Parameter-Efficient Transfer Learning for NLP. ICML 2019

---

### Official Review · Reviewer_EtWw · 2021-10-29

**Correctness:** 4
**Technical Novelty And Significance:** 3
**Empirical Novelty And Significance:** 3
**Recommendation:** 8
**Confidence:** 3

**Main Review:**

Strengths:
- A new perspective of parameter-efficient methods within a unified framework.
- A number of controlled studies which give insights into which components are meaningful and in which cases.
- The released code base can help boost future research in this area.
- Clear and well-written.

Weaknesses:
- No major weaknesses identified.

Questions:
- It would be interesting to see the variance observed when applying parameter-efficient methods. Previous work found large variance in downstream performance across different seeds when performing full fine-tuning [1,2,3]. This is something you could add, for example, to Table 2.

Typos:
- In Figure 2, the performance of Adapter is 21.00, not 20.46
- The "v" subscript is missing in Eq. 5

[1] Dodge et al. Fine-tuning pretrained language models: Weight initializations, data orders, and early stopping. 2021
[2] Mosbach et al. On the Stability of Fine-tuning BERT: Misconceptions, Explanations, and Strong Baselines. ICLR 2021
[3] Bugliarello et al. Multimodal Pretraining Unmasked: A Meta-Analysis and a Unified Framework of Vision-and-Language BERTs. TACL 2021

**Summary Of The Paper:**

This paper investigates recent parameter-efficient methods that have been shown to achieve good performance in a variety of NLP tasks. The authors analyse their connections and propose a unified framework which subsumes a number of existing approaches. The authors then compare their performance across four NLP tasks, with varying level of difficulty and amount of available resources. Here, compared to full fine-tuning, they find that parameter-efficient approaches perform well on simpler tasks (MNLI and SST2), while showing larger gaps on more challenging tasks (XSum and MT). A series of controlled experiments centred around the proposed framework shows that parallel adapters applied to the FFN module of a Transformer generally perform better. Applying them to the attention module leads, however, to better gains when the number of additional parameters is small. Finally, the authors combine these findings into a novel MAM adapter module that leads to further gains in every task.

**Summary Of The Review:**

This paper provides a unified framework for parameter-efficient NLP, an important task when serving models at scale. The experiments provide new insights into the performance of existing methods, that, when combined, result in further performance improvements. I think this is a solid paper and recommend its acceptance.

---

> ### Author Response · Authors · 2021-11-15
> **We have added standard deviation numbers in the revision**
>
> Thank you for your time and encouraging review! We address your questions below:
>
> **-Q1: Variance of parameter-efficient tuning methods**
>
> Thank you for the suggestion! We have added standard deviation numbers on MNLI and SST2 to Table 2 in the revision. MNLI and SST2 are relatively high-resource compared to other GLUE datasets, and the standard deviation on them is not large as shown in Table 2. [1] has reported similar standard deviation numbers on MNLI and SST2.
>
> For high-resource machine translation and summarization tasks such as en-ro and XSum, we have rerun the most performant methods in Table 6 with 3 different random seeds and reported the mean/standard deviation in Table 6, please check the revision for the update – the variance is quite small in XSum/en-ro.
>
> **-Q2: Typos**
>
> Thanks for catching those typos! We have fixed them in the revision.
>
>
> [1] Hu et al. LoRA: Low-rank adaptation of large language models. Preprint 2021

---

### Official Review · Reviewer_JZ1E · 2021-11-02

**Correctness:** 4
**Technical Novelty And Significance:** 4
**Empirical Novelty And Significance:** 4
**Recommendation:** 10
**Confidence:** 5

**Main Review:**

Strengths:

1. The analysis and conclusion on Adapters, Prefix Tuning, and LoRA are very penetrating. Specifically, the paper derives an equivalent form of preﬁx tuning to establish its connection with adapters in a unified view.


2. The paper proposes a uniﬁed framework for parameter-efﬁcent tuning that includes several state-of-the-art methods as instantiations.

3. The paper is well organized and well-motivated with theoretical analysis and proof.

4. Experiment setting and analysis are comprehensive; the proposed method Scaled PA consistently shows its advantages against the other baselines.


**Summary Of The Paper:**

This paper breaks down the design of state-of-the-art parameter efﬁcient transfer learning methods and presents a uniﬁed framework that establishes connections between them. The paper re-frames them as modifications to specific hidden states in pre-trained models and defines a set of design dimensions along which different methods vary, such as the function to compute the modification and the position to apply the modification. Experiments on machine translation, text summarization, language understanding, and text classification have been conducted, which indicates that the unified framework enables can instantiate new parameter-efficient fine-tuning methods that tune fewer parameters.

**Summary Of The Review:**

Overall, this paper is well-written and well-motivated. I think this paper attracts lots of researchers and will inspire future works in parameter efficient training.

---

> ### Author Response · Authors · 2021-11-15
> **Thank you!**
>
> Thank you for your time and encouraging comments!

---

### Public Comment · ~Zhiqiang_Shen1 · 2022-02-21
**Congrats! Would like to introduce a similar paper in the vision domain regarding partial fine-tuning paradigm.**

Dear Authors,

Congratulations on the acceptance of the paper and great job!

I found the idea of parameter-efficient fine-tuning in NLP is somewhat similar to the partial fine-tuning strategy in the computer vision domain, so I'd like to mention a related paper which also comes from Carnegie Mellon University, in case any vision guys are interested in this kind of approach, i.e., the partial or parameter-efficient fine-tuning strategy.

"Partial Is Better Than All: Revisiting Fine-tuning Strategy for Few-shot Learning" (AAAI 2021) (https://arxiv.org/abs/2102.03983)

---

### Decision · Program_Chairs · 2022-01-20

**Decision:**

Accept (Spotlight)

**Comment:**

The paper reviews and draws connections between several parameter-efficient fine-tuning methods.

All reviewers found the paper addresses an important research problem, and the theoretical justification and empirical analyses are convincing.